# Highly Sensitive and Highly Emissive Luminescent Thermometers for Elevated Temperatures Based on Lanthanide-Doped Polymers

Liubov Tcelykh [1], Egor Latipov [2], Leonid Lepnev [3], Andrei Anosov [4], Vladislava Kozhevnikova [1], Natalia Kuzmina [1] and Valentina V. Utochnikova [1,4,*]

1   Chemistry Department, M.V. Lomonosov Moscow State University, 1/3 Leninskie Gory,
    119991 Moscow, Russia
2   Institute of Nanotechnology of Microelectronics of the Russian Academy of Sciences, Leninskiy Prospekt, 32A,
    119334 Moscow, Russia
3   P.N. Lebedev Physical Institute of the Russian Academy of Sciences, Leninsky pros. 53,
    119992 Moscow, Russia
4   Materials Science Department, M.V. Lomonosov Moscow State University, 1/58 Leninskie Gory,
    119991 Moscow, Russia
*   Correspondence: valentina.utochnikova@gmail.com

**Abstract:** Lanthanide coordination compounds contining multiple lanthanides are the most promising candidate materials for luminescent thermometry. Sensing elevated temperatures requires highly stable complexes and matrices, such as those of thermally stable polymers. However, most high-temperature polymers are not optically inert, and this can affect their thermometric properties, including decreasing their intensity and sensitivity. In the present paper, the proper selection of the combination of a matrix and two emitters allowed us to obtain a highly sensitive and highly emissive luminescent thermometry material, $1\{5[Tb(Bz)_3Phen]_2+1[Eu(Bz)_3Phen]_2\}:4PI4050$, based on terbium and europium complexes in poly(ethylene glycol) diacrylate (PI4050), which is suitable for the detection of temperatures up to 200 °C.

**Keywords:** luminescent thermometry; lanthanide complexes; aromatic carboxylates; terbium benzoate; europium benzoate; PI4050; PI2050; PI4072

## 1. Introduction

Luminescence thermometry combines high accuracy with the versatility of the application [1–8], making it the most promising solution for measuring the temperature of small, fast-moving, sensitive, or hard-to-reach objects. Luminescence thermometry in the high-temperature range is the most ambitious application [9], and usually requires temperature mapping. High-temperature luminescent thermometers are particularly important, as they are essential tools for non-contact temperature measurement in high-temperature environments, where traditional contact-based temperature measurement techniques may not be feasible or safe. They are typically made using a combination of luminescent materials, host materials, and optical fibers or other components that can withstand high temperatures and maintain good performance over time [10,11].

Coordination compounds of terbium and europium, which combine narrow emission bands with a constant position and high luminescence intensity, are the most promising compounds [12–16]; combined within one material, they can enable the use of the luminescence intensity ratio (LIR) as a temperature-dependent luminescence parameter, meaning these materials are not subjected to additional calibrations.

Due to their lower thermal stability, lanthanide coordination compounds are not usually used for luminescent thermometry at elevated temperatures, except in several of our recent works [17–20]. We showed that the mixture of complexes demonstrates higher

sensitivity than one bimetallic complex, and polystyrene was used as a matrix for thin film deposition [18].

At the same time, despite its high thermal stability and high melting point, polystyrene softens at temperatures over 160 °C, hampering the high-temperature application of composite materials based on it. Another problem that we faced is that the thermal stability of the compound, when subjected to simultaneous UV exposure, significantly decreases. So, in [20], the obtained material only demonstrated reproducibility up to 120 °C, while its thermal stability, according to TGA, exceeded 400 °C. Therefore, the goal of the present work was to select a better polymer host, as well as to study the reproducibility of the obtained systems.

As the objects of study, a mixture of terbium and europium complexes [Tb(Bz)$_3$Phen]$_2$ and [Eu(Bz)$_3$Phen]$_2$ (HBz = benzoic acid, Phen = *o*-phenanthroline) [18,21], doped into one of the polymers from Figure 1, was selected. These polyimide matrices were selected for their well-known high thermal stability [22].

**Figure 1.** Formulae of the organic compounds used in this work. Ligands: (**a**) benzoic acid, (**b**) o-phenanthroline; polymers: (**c**) PI4050, (**d**) PI2050, (**e**) PI4072; photoinitiator: (**f**) Irgacure 369 (2-Benzyl-2-dimethylamino-1-(4-morpholinophenyl)-butanone-1).

## 2. Results and Discussion

Despite lanthanide aromatic carboxylates belonging to the most stable lanthanide-based coordination compounds [23], simultaneous exposure to thermal treatment and UV light causes their decomposition. Thus, it was very important not to overheat the system during our experiment, and to ensure high reproducibility, a particular measurement setup was built (Figure 2).

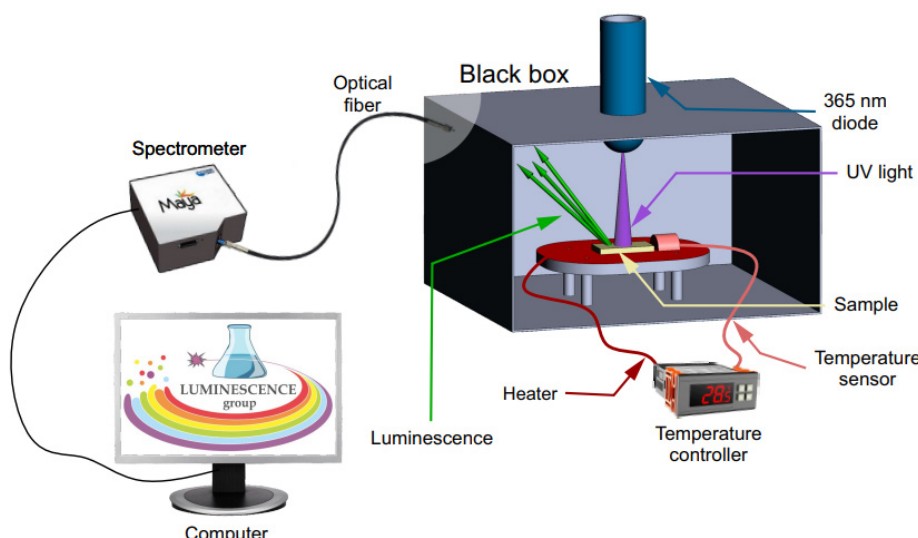

**Figure 2.** Scheme of the built setup.

## 2.1. Measurement Setup Construction

The temperature was set using a PTC ceramic heater with a 230 °C cut-off temperature to prevent the sample from overheating, which may cause decomposition and degradation. An MH1301 B controller with a Pt100 resistance thermometer was used to perform temperature control. The heater was placed in a custom dark box, equipped with an optifiber light guide. The optifiber was connected to the Maya 2000 Pro spectrometer, which was connected to a PC, without a collimator or light filters. An LED with a 365 nm wavelength (100 mW) was used as a luminescence excitation source.

## 2.2. Synthesis and Characterization

[Ln(Bz)$_3$Phen]$_2$ (Ln = Tb, Eu) was obtained and characterized as in [18,24]. It was isostructural to the previously published dimeric terbium complex [Tb(Bz)$_3$Phen]$_2$ (CCDC identifier SAJGEQ), containing two terbium atoms, bound with the two $\mu_2$:$\kappa^2$–$\kappa^1$ and two $\mu_2$:$\kappa^1$–$\kappa^1$ benzoate ligands, while two other benzoate ligands and two phenanthroline ligands adopted a $\kappa^2$ coordination mode. Composite materials were obtained by adding the powders of the coordination compounds, in mass ratios of [Ln(Bz)$_3$Phen]$_2$:PI = 1:4 and [Tb(Bz)$_3$Phen]$_2$:1[Eu(Bz)$_3$Phen]$_2$ = 5:1 (or 0.83:0.17 with respect to the matrix), into the photo-cured resin, followed by the addition of the photoinitiator (less than 5% of the mass of the matrix). Photopolymerization was carried out via exposure to light, with a wavelength of 365 nm, for 10 min, and between two cover glasses to exclude air access. The obtained sample assignment is given in Table 1.

**Table 1.** Composite assignment.

| | |
|---|---|
| **LTPI1** | 1{5[Tb(Bz)$_3$Phen]$_2$+1[Eu(Bz)$_3$Phen]$_2$}:4PI4050 |
| **LTPI2** | 1{5[Tb(Bz)$_3$Phen]$_2$+1[Eu(Bz)$_3$Phen]$_2$}:4PI2050 |
| **LTPI3** | 1{5[Tb(Bz)$_3$Phen]$_2$+1[Eu(Bz)$_3$Phen]$_2$}:4PI4072 |

The FTIR spectra of PI4050, PI2050, and PI4072, as well as **LTPI1-3** (Figure 3), demonstrate the absence of coordinated solvent molecules, which is significant for elevated temperature luminescence thermometry; if present, solvent molecules could be irreversibly removed upon heating, affecting the luminescence properties. Moreover, the spectra of the composite materials are superpositions of the complex and polymer spectra with a predominance of the bands of polymers, which is obvious, given its mass fraction. These data confirm the absence of any chemical reaction between components of the composites.

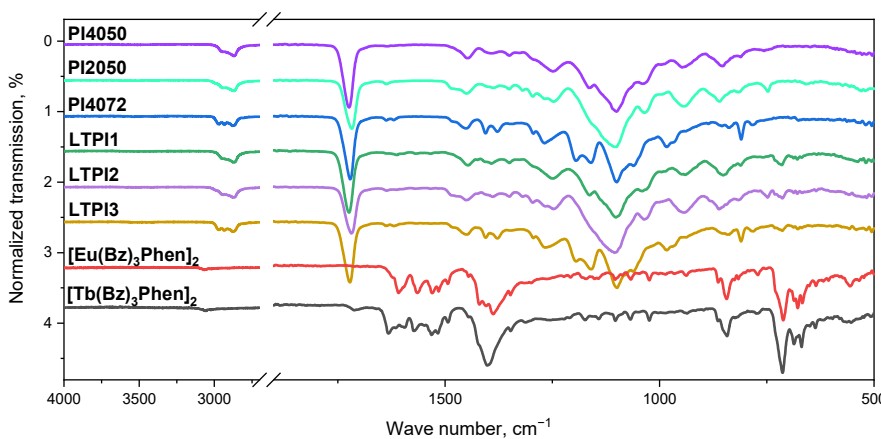

**Figure 3.** Infrared spectroscopy data.

The obtained coordination compounds are thermally stable up to 250 °C (Figure S2), the matrices are stable up to 200 (PI2050 and PI4070) and 250 °C (PI4050), and the obtained composites are thermally stable up to at least 200 °C (Figures 4, S4 and S5), which satisfies the requirements of elevated temperature thermometry.

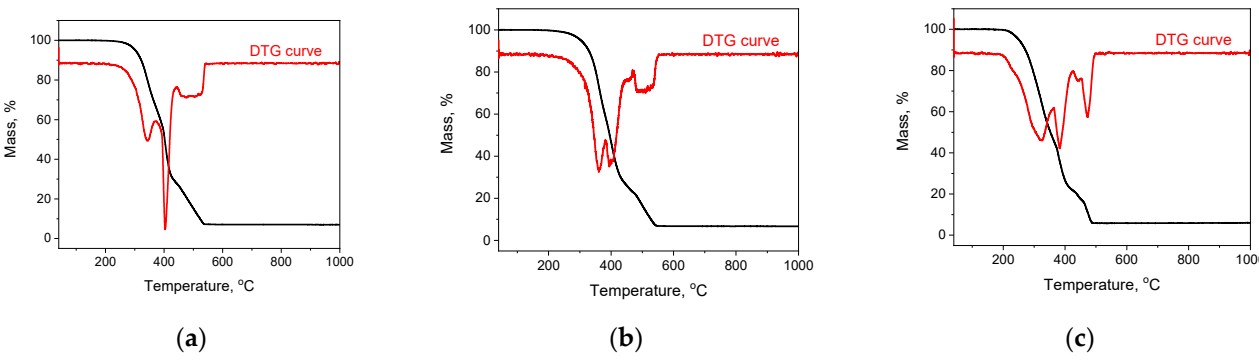

(**a**)  (**b**)  (**c**)

**Figure 4.** Thermal analysis data (10 °C/min) for (**a**) **LTPI1**, (**b**) **LTPI2**, and (**c**) **LTPI3**.

The morphologies of all the samples were studied, and are presented in Figure 5 and ESI (Figure S6). These data demonstrate the acquisition of smooth samples. The SEM photographs show powder grains in the complexes, since the complexes are insoluble in the matrix and do not react with it. In this case, the powders and matrices are thoroughly mixed, which ensures a uniform distribution of the powder complexes in the matrix.

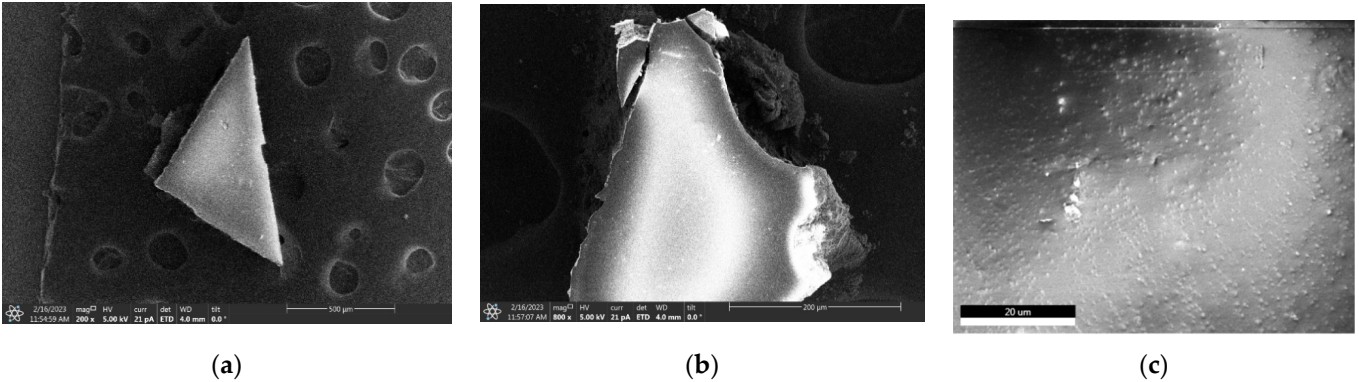

(**a**)  (**b**)  (**c**)

**Figure 5.** SEM data for (**a**) **LTPI1**; (**b**) **LTPI2**, and (**c**) **LTPI2**.

## 2.3. Luminescence

### 2.3.1. Room Temperature Data

Before further investigation, [Tb(Bz)$_3$Phen]$_2$ and [Eu(Bz)$_3$Phen]$_2$ was doped separately into each of the polymers to ensure that their luminescence was not quenched by the selected materials (Figure 6). This was followed by the acquisition of the **LTPI1-3** composite, which also demonstrate intense luminescence, and consist of typical narrow-band emission of both terbium (centered at 545 nm) and europium (centered at 613 nm). The excitation spectra (see ESI Figure S7), recorded at the emission of both ions, consist of a broad band of through-ligand excitation. In contrast, the spectra, recorded upon the emission of europium, also contains a narrow band, centered at 390 nm, corresponding to direct europium excitation ($^7F_{0,1} \rightarrow ^5L_6$).

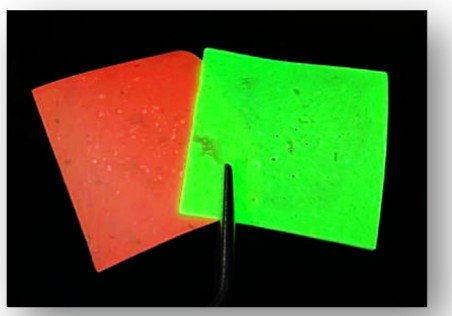

**Figure 6.** Photoluminescence of [Tb(Bz)$_3$Phen]$_2$ and [Eu(Bz)$_3$Phen]$_2$, doped into PI4050.

The luminescence spectra of LTPI1-3 composite materials (Figure 7) contain high-intensity bands of both terbium and europium ions [25,26]. In this case, we chose a ratio of terbium and europium complexes of [Tb(Bz)$_3$Phen]$_2$:[Eu(Bz)$_3$Phen]$_2$ = 5:1, wherein the bands of terbium ($^5D_4$-$^7F_5$) and europium ($^5D_0$-$^7F_2$) were comparable in intensity. This was necessary, since the calculation of temperature sensitivity in such systems is carried out according to $LIR = \frac{I(at\ 543\ nm)}{I(at\ 613\ nm)}$.

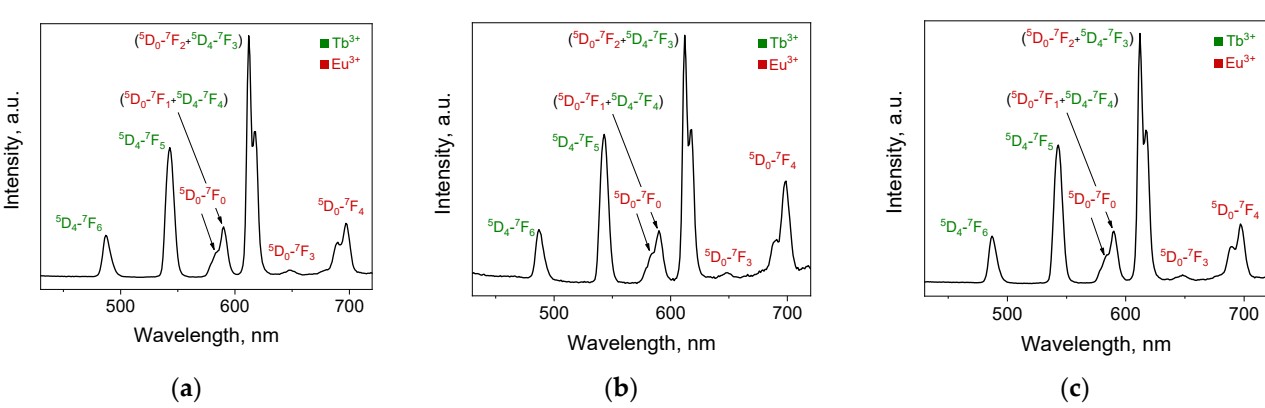

**Figure 7.** Luminescence spectra ($\lambda_{ex}$ = 350 nm) of (**a**) **LTPI1**, (**b**) **LTPI2**, and (**c**) **LTPI3**.

The lifetimes of the excited states ($\tau$) were slightly affected by the doping and depended on the selected polymer (Table 2), as well as the luminescence quantum yields (PLQYs). The lowest value of 7% is observed for **LTPI2**, while the remaining two composites demonstrate rather high quantum yields of 22% and 27%. Based on these data, **LTPI1** was selected to study luminescent thermometry.

**Table 2.** Luminescent data for [Tb(Bz)$_3$Phen]$_2$, [Eu(Bz)$_3$Phen]$_2$, **LTPI1**, **LTPI2**, and **LTPI3**.

| Sample | PLQY, $\pm 5\%$ | $\tau$(Tb), $\pm 0.01$ ms | $\tau$(Eu), $\pm 0.1$ ms |
|---|---|---|---|
| [Tb(Bz)$_3$Phen]$_2$ | 14 | 0.13 | - |
| [Eu(Bz)$_3$Phen]$_2$ | 99 | - | 1.24 |
| LTPI1 | 22 | 0.20 | 1.25 |
| LTPI2 | 7 | 0.20 | 1.51 |
| LTPI3 | 27 | 0.22 | 1.70 |

### 2.3.2. Luminescent Thermometry

Temperature-dependent luminescence was initially studied in the whole range using the constructed setup (20–220 °C). It was ensured that the emission of both metals was present across this whole range. No softening of the materials was observed during measurements. Additionally, to exclude any phase transition processes, temperature-dependent PXRD data were recorded for [Eu(Bz)$_3$Phen]$_2$ powder in the range of 30–120 °C (Figure 8b and ESI). A special accessory was constructed to enable temperature-dependent PXRD measurements (scheme in Figure 8a). These data reveal that the crystal structure is preserved, while the cell volume obviously increases upon heating (see ESI for indexing details).

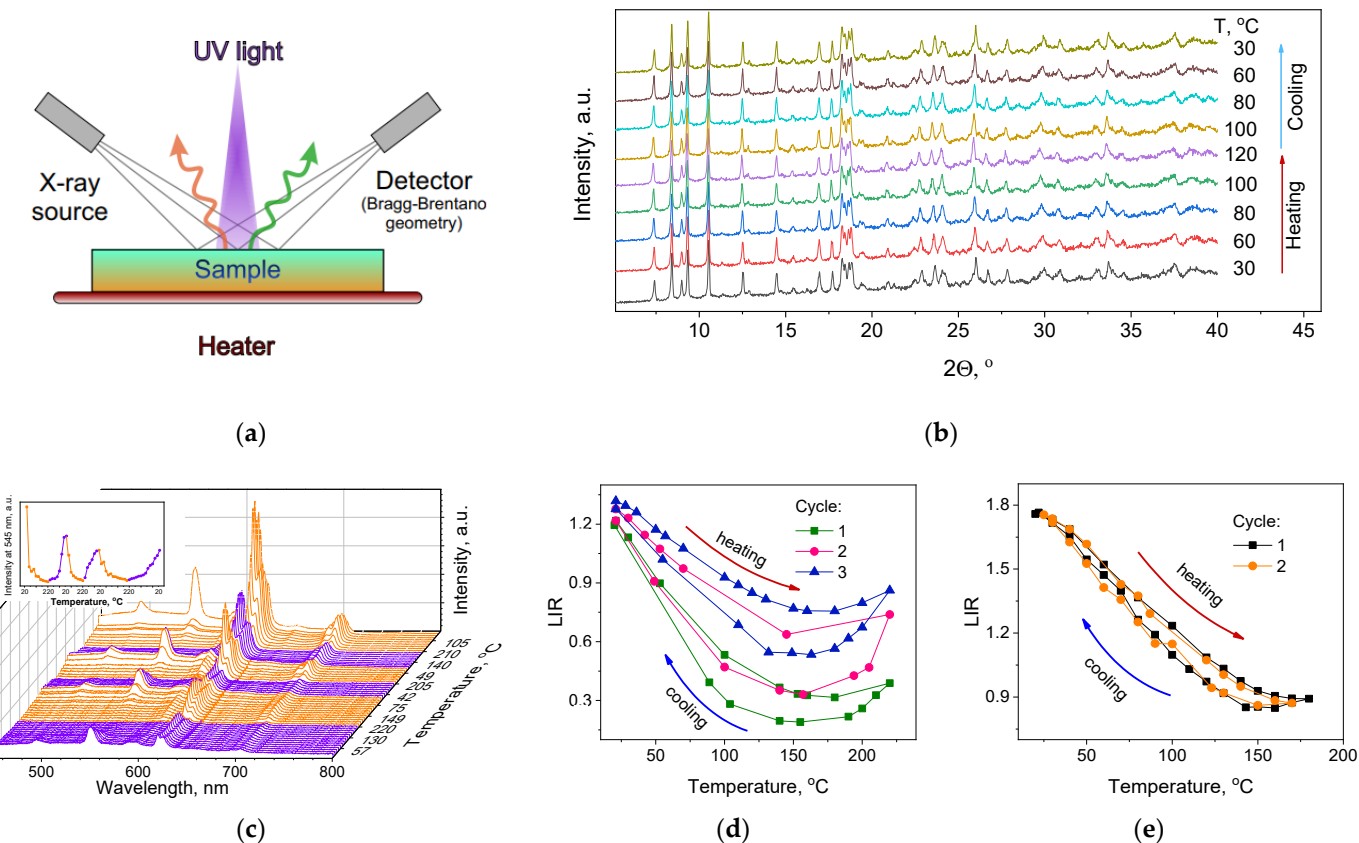

**Figure 8.** (**a**) Scheme of the setup for PXDR measurement with heating and (**b**) PXDR data for [Eu(Bz)$_3$Phen]$_2$ at different temperatures. Temperature-dependent luminescence of **LTPI1**: (**c**) spectra ($\lambda_{ex}$ = 365 nm); inset shows heating profile: violet color corresponds to heating, while orange corresponds to cooling. (**d,e**) LIR in the temperature ranges of (**d**) 20–220 °C and (**e**) 20–180 °C.

However, according to our study of reproducibility, the sample was subjected to degradation during the measurement, even though it was thermally stable at this temperature range according to the TGA data (Figure 8c). Such a result can be associated with simultaneous exposure to UV radiation when the sample was heated, or longer holding

of the material at a high temperature when measuring the temperature dependencies of the luminescence spectra. To select the temperature range at which the samples worked reproducibly, a series of heating and cooling experiments was performed at various temperatures (160–220 °C) (see ESI). This established that luminescence intensity was restored completely after cooling only if the sample was heated below 180 °C. So, according to this, **LTPI1** can be used as a thermometer in the temperature range of 20–180 °C.

To determine the uncertainty of the LIR determination, the standard deviation of the intensities was calculated at each temperature, and the standard deviation of the LIR value was calculated from these data. For this purpose, the luminescence spectra were recorded several times at each temperature up to 220 °C (see ESI). These data also support the conclusion that degradation starts upon UV irradiation with simultaneous heating over 180 °C. Indeed, up to this temperature, the luminescence intensity is preserved, while at 190–220 °C, the intensity decreases with every measurement. Additionally, the obtained data demonstrate that the observed "hysteresis" is within the experimental error.

Sensitivity represents the accuracy of temperature measurement at a given temperature:

$$S_r = \frac{1}{LIR} \cdot \frac{dLIR}{dT} \tag{1}$$

It was found to be in the range of 20–180 °C, where it reached 0.85%/°C (Figure 9).

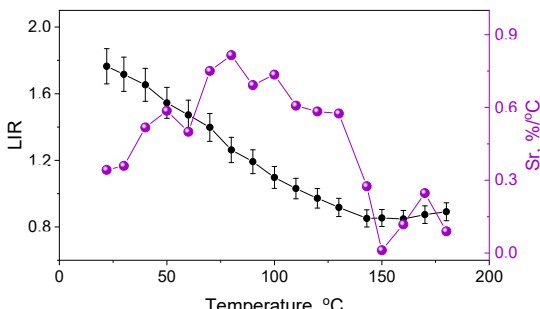

**Figure 9.** Luminescence intensity temperature dependencies and relative sensitivity (Sr) of **LTPI1**.

## 3. Experimental Section

### 3.1. Materials and Methods

All solvents and chemicals (terbium nitrate pentahydrate ($Tb(NO_3)_3 \cdot 6H_2O$, 99.9%), europium chloride hexahydrate ($EuCl_3 \cdot 6H_2O$, >99%), 1,10-phenanthroline (Phen, 99%), benzoic acid (H(Bz)), 99.5%), poly(ethylene glycol) diacrylate (PI4050, QL200), poly(ethylene glycol) dimethacrylate (PI2050, QL200), trimethylolpropane triacrylate (PI4072, QL200), 2-Benzyl-2-dimethylamino-1-(4-morpholinophenyl)-butanone-1 (Irgacure 369, QL100) were purchased from commercial sources. Thermal analysis was carried out using an STA 449 F1 Jupiter thermal analyzer (NETZSCH, Selb, Germany) in a temperature range of 40–1000 °C in air, and at a heating rate of 10° min$^{-1}$. The evolved gases were simultaneously monitored during the TA experiment using a coupled QMS 403 Aëolos Quadro quadrupole mass spectrometer (NETZSCH, Selb, Germany). The mass spectra were registered for the species with the following *m/z* values: 18 (corresponding to $H_2O$), 44 (corresponding to $CO_2$), 45 (corresponding to $C_2H_5OH$), and 31 (corresponding to $CH_3O$). The **IR spectra** were recorded using a Thermo Scientific™ Nicolet™ iS50 FTIR Spectrometer with powders via ATR (Waltham, MA, USA). **Powder X-ray diffraction (PXRD)** was performed using Bruker D8 Advance (λ(Cu-Kα) = 1.5418 Å; Ni filter) with a step size of 0.020°. The patterns were indexed via an SVD index [27] using the TOPAS 4.2 software [28]. Then, the powder patterns were refined using the Pawley method. **Scanning electron microscopy (SEM)** was performed using an FEI Helios G4 CX dual beam scanning electron microscope. Preliminarily, a thin layer of Cr (10 nm) was coated on the samples using SPI Supplies. The **luminescence spectra** were determined and the measurement of **quantum yields** carried out using a FluoroMax (HORIBA) spectrometer at room temperature, excitation was

performed through a ligand, and the absolute method in the integration sphere was used. The registration of **luminescence spectra upon heating** was carried out on a Maya 2000Pro spectrofluorometer (Ocean Optics) using a heating element.

### 3.2. Synthesis

**The synthesis of [Ln(Bz)₃Phen]₂** (Ln = Tb, Eu) was carried out through an exchange reaction between LnCl₃Phen (in situ from a mixture of lanthanide chlorides and Phen) and potassium benzoates (in situ from KOH and Hobs), as in refs. [18,24].

**Composite materials** based on [Ln(Bz)₃Phen]₂ (Ln = Tb, Eu) were prepared by adding the powders of the coordination compounds in the desired proportions into the photo-cured resin, with added photoinitiator (less than 1% by weight), in a ratio of 1 to 4. Photopolymerization was carried out via exposure to light with a wavelength of 365 nm, for 10 min, between two cover glasses.

## 4. Conclusions

We demonstrated that polyimides are suitable as hosts for lanthanide-based materials with thermally dependent luminescence. The luminescence of both terbium and europium was visible and, despite the decrease in the quantum yield due to doping, remained intense. The polyimide hosts did not soften in the temperature range under investigation. At the same time, we demonstrated that the thermal stability of the compounds decreased with simultaneous exposure to UV light; therefore, studying novel materials for high-temperature luminescence thermometry is particularly important for analyzing their reproducibility. As a result, we obtained a composite material based on a mixture of terbium and europium benzoates with phenanthroline in poly(ethylene glycol) diacrylate (PI4050), which demonstrated high thermal stability and luminescent thermometry properties up to 180 °C. Particular attention was paid to reproducibility, as the simultaneous exposure to UV light and heating resulted in a loss of reproducibility up to >180 °C. These results are promising for future research, and show that by choosing a thermally stable and optically transparent matrix, one can obtain an efficient fluorescent thermometer for high temperature measurements.

**Supplementary Materials:** The supporting information can be downloaded at: https://www.mdpi.com/article/10.3390/inorganics11050189/s1. Figure S1: PXDR data of (1) Tb(Bz)3Phen; (2) Eu(Bz)3Phen; (3) theoretical PXDR pattern of Tb(Bz)3Phen calculated from the crystal structure (CCDC identifier SAJGEQ); Figure S2: Thermal analysis with mass-detection of the evolved gases of (a) Tb(Bz)3Phen, (b) Eu(Bz)3Phen; Figure S3: Thermal analysis (10 °C/min) of polymers: (a) PI4050, (b) PI2050, (c) PI4072 and (d) PI4050, PI2050, PI4072; Figure S4: Thermal analysis of composite and polymer (a) LTPI1 and PI4050, (b) LTPI2 and PI2050, (c) LTPI3 and PI4072; Figure S5: Thermal analysis (5 C/min) of LTPI1, PI4050, Tb(Bz)3Phen, Eu(Bz)3Phen; Figure S6: SEM data of (a,b) LTPI1, (c,d) LTPI2, (e,f) LTPI3; Figure S7: Excitation spectra of (a) LTPI1 ($\lambda_{em}$ = 543 nm), (b) LTPI1 ($\lambda_{em}$ = 612 nm), (c) LTPI12 ($\lambda_{em}$ = 543 nm), (d) LTPI2 ($\lambda_{em}$ = 612 nm), (e) LTPI13 ($\lambda_{em}$ = 543 nm), (f) LTPI13 ($\lambda_{em}$ = 612 nm); Figure S8: Lifetimes of (a) Tb(Bz)3Phen at 487 nm; (b) Tb(Bz)3Phen at 543 nm, (c) Eu(Bz)3Phen at 612 nm, (d) Eu(Bz)3Phen at 697 nm, (e) LTPI1 at 487 nm, (f) LTPI1 at 543 nm, (g) LTPI1 at 612 nm, (h) LTPI1 at 697 nm, (i) LTPI2 at 487 nm, (j) LTPI2 at 543 nm, (k) LTPI2 at 612 nm, (l) LTPI2 at 697 nm, (m) LTPI3 at 487 nm, (n) LTPI3 at 543 nm, (o) LTPI3 at 612 nm, (p) LTPI3 at 697 nm ($\lambda$ex = 350 nm); Figure S9: Reproducibility of LTPI1. Dependence of luminescence spectra (a,c,e,g,i) and LIR (b,d,f,h,j) on temperature when heated to 130 °C (a,b), 160 °C (c,d), 180 °C (e,f), 200 °C (g,h) and 220 °C (i,j). ($\lambda$ex = 365 nm); Figure S10: Luminescence spectra of LTPI1 measured several times successively in the range from 50 to 220 °C in steps of 10 degrees ($\lambda$ex = 365 nm); Figure S11: PXRD data for Eu(Bz)3Phen at different temperatures: (a) 30_start, (b) 60_heat, (c) 80_heat, (d) 100_heat, (e) 120_heat, (f) 100_cool, (g) 80_cool, (h) 60_cool, (i) 30_finish. Experimental curve (blue), fitting curve (red), difference (grey).

**Author Contributions:** Conceptualization, V.V.U.; methodology, L.T.; software, A.A., L.L. and E.L.; validation, L.L., E.L. and V.K.; formal analysis, E.L. and L.T.; investigation, L.T.; resources, L.L., E.L. and A.A.; data curation, L.T.; writing—original draft preparation, V.V.U.; writing—review and editing,

V.V.U. and L.T.; visualization, A.A. and L.T.; supervision, V.V.U. and N.K.; funding acquisition, V.V.U. All authors have read and agreed to the published version of the manuscript.

**Funding:** This research was funded by the Russian Science Foundation (grant number 20-73-10053).

**Data Availability Statement:** The data are available upon request.

**Acknowledgments:** We thank M. Vyaltsev and V. Zharenova, who participated in the early stage of this work. We are also extremely thankful for Alexander Goloveshkin's contribution to the XRD study in this work.

**Conflicts of Interest:** The authors declare no conflict of interest.

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
