# Peer review of "Highly Sensitive and Highly Emissive Luminescent Thermometers for Elevated Temperatures Based on Lanthanide-Doped Polymers"

_inorganics, doi:10.3390/inorganics11050189_

Round 1

Reviewer 1 Report

In this manuscript, the authors used a mixture of terbium and europium complexes Tb(Bz)3Phen and Eu(Bz)3Phen doped into polyimide matrices with high thermal stability for luminescent thermometers. The paper should be revised before publications.

1.   The abstract is not clear to the readers. The authors only indicated that “the proper selection of the combination of matrix and two emitters allowed obtaining a highly sensitive and highly emitting luminescent thermometry material, suitable for the detection of temperatures up to 200 °C”. The authors should give more detailed information concerning the method used in this work.

2.   The component ratios in the Table 1 are not clear. Is the ratio of Tb complex to Eu complex a molar ratio or weight ratio? In Table 1, for instance, 1[5Tb(Bz)3Phen+1Eu(Bz)3Phen]:4PI4050, here which kind of ratio is for “1:4”?

3.   Are the composites homogenous or not? From Figure 5, it seems that the materials are not homogenous, for instance, LTPI3 material. As you indicated that “LTPI1 can be used as a thermometer in the temperature range of 20-180 °C.”, the SEM of LTPI1 should appear in the main text.

4.   Is there an energy transfer from Tb to Eu in the mixture of these two complexes in the composites?

5.   In Table 2, the PLQY is so high for Eu(Bz)3Phen (99±5%), even can be more than 100%. Please check these data. In addition, for lifetime measurements, the error (±0.1 ms) is too big, you should measure the decay curves slowly. Otherwise, the lifetime (0.13 ms) for Tb complex is not accurate.

6.   For the two complexes, which ref did the authors follow? Ref 8 or 15, or both? As the authors gave different descriptions in the paper.    

Author Response

1. The abstract is not clear to the readers. The authors only indicated that “the proper selection of the combination of matrix and two emitters allowed obtaining a highly sensitive and highly emitting luminescent thermometry material, suitable for the detection of temperatures up to 200 °C”. The authors should give more detailed information concerning the method used in this work.
Thank you.
We have expanded the introduction with the phrase «material 1{5[Tb(Bz)3Phen]2+1[Eu(Bz)3Phen]2}:4PI4050 based on terbium and europium complexes in poly(ethylene glycol) diacrylate (PI4050)».
2. The component ratios in the Table 1 are not clear. Is the ratio of Tb complex to Eu complex a molar ratio or weight ratio? In Table 1, for instance, 1[5Tb(Bz)3Phen+1Eu(Bz)3Phen]:4PI4050, here which kind of ratio is for “1:4”?
Thank you, we have added an explanation to the text: «Composite materials were obtained by adding the powders of coordination compounds in mass ratio [Ln(Bz)3Phen]2:PI = 1:4 and [Tb(Bz)3Phen]2:1[Eu(Bz)3Phen]2 = 5:1 (or 0.83:0.17 with respect to the matrix) into the photo-cured resin, followed by the photoinitiator addition».
3. Are the composites homogenous or not? From Figure 5, it seems that the materials are not homogenous, for instance, LTPI3 material. As you indicated that “LTPI1 can be used as a thermometer in the temperature range of 20-180 °C.”, the SEM of LTPI1 should appear in the main text.
Thank you, we have added SEM of LTPI1 to the figure 5.
4. Is there an energy transfer from Tb to Eu in the mixture of these two complexes in the composites?
Thank you for this question! Based on our previous finding (ref [18] in the paper), we use a mixture of powders of terbium and europium complexes rather than bimetallic compounds, due to the absence of the energy transfer resulted in the increase of sensitivity. Indeed, when mixed with the matrix, the complexes do not dissolve, so we believe that the distance between the particles is so large that it does not allow energy transfer between terbium and europium ions.
At the same time, it was not directly proven in the paper, and we added the corresponding experiment as the answer to your question. For that, we recorded the excitation spectra of the LTPI1 composite in two zones 200-400 and 400-590 nm when recording the spectrum at 613 nm and 543 nm. The bands corresponding to excitation through the terbium ion are present in the spectrum when recorded at 543 nm and are extremely low-intensity when the excitation spectrum is recorded at 613 nm. This indicates that there is no energy transfer between terbium and europium ions or is present in a very small amount when particles of the terbium and europium complex come into contact.

5. In Table 2, the PLQY is so high for Eu(Bz)3Phen (99±5%), even can be more than 100%. Please check these data. In addition, for lifetime measurements, the error (±0.1 ms) is too big, you should measure the decay curves slowly. Otherwise, the lifetime (0.13 ms) for Tb complex is not accurate.
We wrote “±5%” due to the error of the method is 5%, while of course, the value of the quantum yield can not exceed 100%. So the measured value of 99% means that the quantum yield can be equal to 94% to 100% taking into account the error, wince the measured value of the quantum yield is 99%.
For the lifetime measurements, the error is 0.01 ms. It was our typo, thank you for reading carefully. 0.01 is an upper estimate for the total error of the instrument, experiment, and data approximation. Now we have corrected.
6. For the two complexes, which ref did the authors follow? Ref 8 or 15, or both? As the authors gave different descriptions in the paper.  
Thank you for such a careful reading of our work. These are really two different articles with the same synthesis technique. Article [15 (now it is 24)] describes the synthesis of lanthanide carboxylates with phenanthroline, and article [8 (now it is 18)] is about exactly the same complexes as in our article. We have added both links at every mention of synthesis.

Reviewer 2 Report

The scientific content of the ms is interesting and this work thus deserves – according to my opinion – acceptance and publication in INORGANICS. I am sure that the paper will attract the interest of scientists working in the general area of luminescence thermometry, a currently “hot” topic in material and inorganic chemistry. Also, I do believe that the article will receive a respectable number of citations in the future. Salient features of this work – which support my proposal for acceptance – are: (a) Polyimides are appropriate hosts for lanthanoid materials with thermally dependent luminescence. (b) Despite the decrease of the quantum yield because of the Tb(III)/Eu(III) doping, the luminescence remains bright; and (c) The polymeric hosts do not undergo softening in the temperature range studied. The ms is nicely written and well organized. The quality of figures is high and the Supplementary Info is helpful and diagnostic, a model for authors.

Based on the above mentioned, I am more than happy to recommend acceptance of this fine piece of research in INORGANICS. Minor points/comments/suggestions to be taken into account by the authors:

(1)     The formulae of the complexes should be written within square brackets.

(2)     The “Conclusions” section should be enlarged, with emphasis on the perspectives of the present work.

(3)     The authors should briefly discuss the previously published single-crystal X-ray structures of [Ln(Bz)3(Phen)]. Although, I am not sure, I can remember that these complexes are dimeric, i.e. [Ln2(Bz)6(Phen)2]. If yes, the formulae should be written in the correct manner throughout the text.

(4)     Some key references in the area of luminescence thermometry are missing, e.g. the published papers by Murugesu’s and Tangoulis’ groups.

(5)     “Abstract”: The formula of the complexes and the lanthanoids  studies should be provided. Writing “Bimetallic lanthanide coordination compounds, …” is too general.

Author Response

Based on the above mentioned, I am more than happy to recommend acceptance of this fine piece of research in INORGANICS. Minor points/comments/suggestions to be taken into account by the authors:

Thank you for you kind attitude to our paper!
1. The formulae of the complexes should be written within square brackets.
We agree and have replaced all formulas with [Ln(Bz)3Phen]2, considering that that these complexes are dimeric.
2. The “Conclusions” section should be enlarged, with emphasis on the perspectives of the present work.
Thank you, we have enlarged the conclusion:
«As a result, we obtained a composite material, based on a mixture of terbium and europium benzoates with phenanthroline in poly(ethylene glycol) diacrylate (PI4050), which demonstrated high thermal stability and luminescent thermometry properties in the range of up to 180 oC. Particular attention was paid to the reproducibility, as the simultaneous exposure to the UV light and heating resulted in the loss of reproducilibty in the >180 oC temperature range. These results are promising for future research and show that by choosing a thermally stable and optically transparent matrix, one can obtain an efficient fluorescent thermometer for high temperature measurements».
3. The authors should briefly discuss the previously published single-crystal X-ray structures of [Ln(Bz)3(Phen)]. Although, I am not sure, I can remember that these complexes are dimeric, i.e. [Ln2(Bz)6(Phen)2]. If yes, the formulae should be written in the correct manner throughout the text.
The reviewer is absolutely right. we used formula unit Ln(Bz)3(Phen) for brevity, now we have replaced all the formulas by [Ln(Bz)3Phen]2. We added the short description:
“They were isostrutural to the previously published dimeric terbium complex [Tb(Bz)3Phen]2 (CCDC identifier SAJGEQ), containing two terbium atoms, bound with the two μ2:κ2–κ1 and two μ2:κ1–κ1 benzoate ligands, while two other benzoate ligands and two phenanthroline ligands adopted κ2 coordination mode.”
4. Some key references in the area of luminescence thermometry are missing, e.g. the published papers by Murugesu’s and Tangoulis’ groups.
Thank you! We have added new links, including those of these authors, for a more complete overview of current research on the topic of luminescent thermometry.
5. “Abstract”: The formula of the complexes and the lanthanoids studies should be provided. Writing “Bimetallic lanthanide coordination compounds, …” is too general.
Thank you, we have enlarged the abstract for a more detailed explanation of the topic.

Round 2

Reviewer 1 Report

The authors addressed all my concerns, and I think the manuscript can be accepted in present form. However, the text should be checked again to correct something. For instance, "terbium nitrate pentahydrate  (Tb(NO3)3·6H2O, 99.9%)", to make clear Tb(NO3)3·5H2O or Tb(NO3)3·6H2O was used.